



# Case study on the impact of moving broken clouds on the spectral radiance

Jens Duffert[1], Angelika Niedzwiedz[1], Kolja Wagner[1], Juliane Duffert[2], Mario Tobar Foster[3] and Gunther Seckmeyer[1]

[1]Institut für Meteorologie und Klimatologie, Leibniz Universität Hannover, Herrenhäuser Str. 2, D-30419 Hannover, Germany
[2]energy & meteo systems GmbH, Oskar-Homt-Straße 1, 26131 Oldenburg, Germany
[3]Independent Researcher

*Correspondence to*: Jens Duffert (duffert@meteo.uni-hannover.de, jens-duffert@t-online.de)

**Abstract.** This case study shows the possibility of using the Advanced MUltiDIrectional Spectralradiometer (AMUDIS) to simultaneously measure the temporal impact of moving clouds on the spectral radiance in different directions and also the first results. For these aims measurements from January 2024 with the AMUDIS in Hannover (Germany 52.39° N and 9.7° E) for cloudless sky and two moving broken clouds situation with different cloud cover are analysed.

From the measuring range of 380–890 nm and the 140 directions, three wavelengths (480 nm, 530 nm, 585 nm) and three directions were selected. These three directions are in the area between north and east and were measured simultaneously.

From these datasets, the temporal variation was calculated for each case with reference to a cloudless moment in the situation (no clouds were observed in the field of view of the input optics). The variations during a broken cloud situation on 18 January were compared with the variations on 10 January under cloudless conditions. It was determined that the spectral radiance was higher by a factor of up to 3 under cloudy conditions. The results show also that the impact of the clouds is dependent on the wavelength, direction and cloud type. The AMUDIS offers another option for validating radiation transfer models, for example.

## 1 State of art and preliminary work

Clouds have a significant influence on the radiation balance at the Earth's surface (Zelinka et al., 2017; Trenberth et al., 2009). Clouds change their properties at short time intervals, which also causes their optical properties to change. It is therefore necessary to make measurements from different directions, preferably simultaneously (source), in order to determine this influence. Spectral radiance is a suitable measurement variable for this purpose. The spectral radiance (cf. DIN 5031-1, 1982) can be used to derive different radiometric quantities and to describe interaction processes between solar radiation and atmospheric components such as: aerosols (Dubovik and King, 2000), trace gases (Hönninger et al., 2004) and clouds that vary in time and spatial distribution (Hirsch et al, 2012), as well as to estimate and validate cloud - radiation properties for ground based radiation observations (Bernhard and Seckmeyer, 1999; Cordero et al. 2023; Cronin et al., 2006; Turner et al., 2007; Brandau et al., 2010; Shupe et al., 2015; Knist und Russchenberg, 2018; Ebell et al., 2020; Villefranque and Hogan, 2021) and for comparison to the results from radiative transfer modelling (Thomas and Stamnes, 1999; cf. Dubovik and King, 2000;



Oreopoulos and Mlawer 2010; Hirsch et al., 2012). Furthermore, the spectral radiance can be used to estimate the effect of solar radiation on humans (Seckmeyer et al., 2013; Schrempf et al., 2018). Focus on the ground based detection solar radiance can be derived using various measurement systems like:

35 - Instruments based on spectroradiometers which detect the wavelength-dependent radiance within a scan (sky scans) with a double or single monochromator, e.g. for the UV range of the solar spectrum (Wuttke and Seckmeyer, 2006; Weihs et al., 2000). Although spectroradiometers fulfill high measurement standards, they are spatially, temporally and spectrally limited due to the scanning principle. As a limitation, rapidly varying parameters of the atmosphere, such as clouds, cannot be detected with sufficient resolution due to the temporal (days), spatial (each sky segment) 40 and wavelength-dependent (few chosen wavelengths) scanning method of the systems.

- With the charged coupled device (CCD) array spectrometers it became possible to detect the spectral radiance simultaneously over their operational wavelength range (Kouremeti et al., 2008; Dunagan et al., 2011). However, one scan over the whole sky still takes several minutes (e.g. approximately 15 minutes for 150 measurements points) 45 depending on the speed of the positioning device.

- Solar radiation data can also be detected with Hemispherical Sky Imager (HSI) systems. In general it is possible with HSI systems to: detect clouds and calculate the total cloud cover (Kazantzidis et al., 2012; Yamashita et al., 2004), to classify and analyse different cloud types, especially investigating their radiative effects (Calbó and Sabburg, 2008; 50 Cheruy and Aires, 2009; Paliwal and Kumar, 2009; Mellit and Kalogirou, 2008; Tapakis and Charalambides, 2013; Cheng and Yu, 2015; Gan et al., 2017; Heinle et al., 2010; Li et al., 2016; Satilmis et al., 2020). Due to its automated operation, high spatial, temporal and limited spectral resolutions, HSI ground-based instruments are also an essential supplement to visual observation, since they cover the same spectral range as the human eye (Werkmeister et al., 2015).

55

- Non-scanning multidirectional spectroradiometers (multidirectional spectral radiometer (MUDIS), advanced multidirectional spectral radiometer (AMUDIS)) simultaneously detect the direction, wavelength, time and measured quantity (Riechelmann et al., 2013; Seckmeyer et al., 2018; Tobar Foster et. al., 2021). Due to some technical limitations of the MUDIS prototype (Riechelmann et al., 2013) as well as the experience gained with the instrument, 60 a new large-scale device (AMUDIS) was developed (DFG approval: GZ: INST 187 / 555-1 FUGG).

With the AMUDIS, it is possible to reduce the temporal uncertainties in measurements in different direction of the atmospheric variability. Further details regarding the setup and functional principle of the AMUDIS follow in chapter 2.1.

In this publication, it will be shown for the first time that the AMUDIS can be used to detect and quantify temporal variability
of spectral radiance under the influence of clouds (single clouds, broken clouds) compared to a cloudless sky. Measurements
from 10 January 2024 and 18 January 2024 were used for this study.

## 2 Devices, Materials and Methods

### 2.1 AMUDIS

The advanced multidirectional spectroradiometer (AMUDIS) is a novel measuring system, which measures the spectral
radiance in the wavelength range from 280 - 1700 nm in more than 140 sky directions within a few seconds. As already
described in Seckmeyer et al. (2018), Tobar Foster et al. (2021). It is based on the prototype multidirectional spectroradiometer
(MUDIS) (Riechelmann et al. 2013).

The device consists of:

- One weatherproof hemispherical input dome with three optical fibre bundles for each spectral range (UV:280-390
  nm, VIS: 380–890 nm, NIR: 880-1700 nm). These fibres are embedded in the hemispherical dome in more than 140
  sky directions (in total 435 fibres, see figure 1). As characterized by Tobar Foster et al. (2021), each fibre provides
  an individual and wavelength-dependent field of view (FOV).

- A spectrometer with a double monochromator connected to a CCD chip camera for the UV range, and two
  spectrometers with single monochromators connected to one camera each for the VIS (CCD chip) and NIR (InGaAs
  chip) range. The optical fibres for each spectral range are bundled and connected to the respective spectrometer via
  an adapter lined up to a slit.

Radiance is collected by the optical-fibre-based hemispheric input dome multidirectional and simultaneously. After
passing the optical components (collimator mirror, grid, etc.) inside the spectrometers, the light is detected by each camera
sensor. Both spectral and spatial information can be obtained from the acquired images.

The measuring device has been calibrated for spectral radiance both in the laboratory and in field operation, as described
in Niedzwiedz et al. (2021) and (2022). Since the ambient temperature influences the stability and uncertainty of the
measurement data the measuring device is operated in a temperature-controlled laboratory or in a temperature-controlled
box (mean value: +/- 0.5°C (+/- 0.35 °C) /max value: +/- 1.5°C) for outdoor/field measurements, in order to ensure stable
temperature conditions of the measuring device.

For a better understanding and visualization of the analysed cloud situations and the orientation of the analysed fibres in
the upper hemisphere, the different fibre directions were combined with an image of the Hemispherical Sky Imager (HSI)



(see figure 1). The position and FOV of the fibres are based on the measurements described in Tobar Foster et al. (2021). The different fields of view for the three selected fibres are listed in table 1. The HSI system consists of a single-lens reflex camera, a fisheye objective and a temperature control box. The images were taken at a 1-minute intervals with a field of view of 180°. The sky directions given here correspond to the geographical coordinates at the measurement

location. Note that east (left) and west (right) are switched in the images. This means for the analysed fibres that fibre 90 is in the north segment, fibre 28 is shifted to the northeast and fibre 4 is in the east segment (see figure 1).

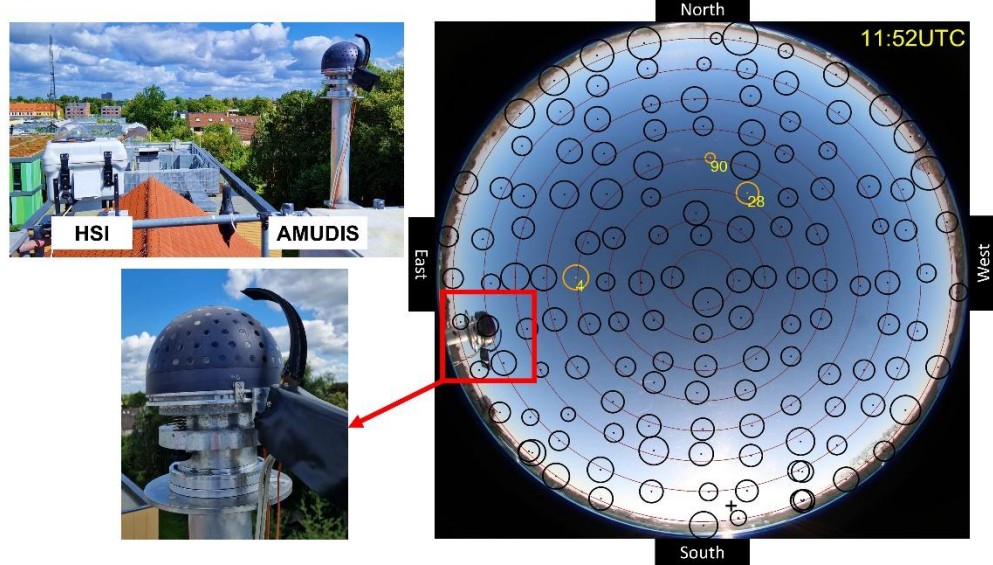


**Figure 1: Left: The figure shows the measurement setup on IMUK's measurement platform. Top: HSI system (left) and the input optics of the AMUDIS. Bottom: The hemispherical input optics under the black weather cover with shadow band. The shadow band covers direct sunlight. Direct sunlight leads to oversaturation of CCD chips as well as scattering effects within the fibres. Right: Combined HSI image with AMUDIS sky directions. Fibre 4, 28, 90 are the sky directions, which are used for data analysis here. The**
**different sizes of the circles are intended to represent the wavelength-dependent field of view, here for the wavelength 480 nm.**

| Fibre 4 | | Fibre 28 | | Fibre 90 | |
|---|---|---|---|---|---|
| 480 nm | 9.0° | 480 nm | 8.0° | 480 nm | 3.8° |
| 530 nm | 8.6° | 530 nm | 7.6° | 530 nm | 3.8° |
| 585 nm | 8.3° | 585 nm | 7.2° | 585 nm | 3.9° |

**Table 1: The FOV (fields of view) are based on the analysis described in Tobar Foster et al. (2021) for fibres 4, 28 and 90. Please note that the FOV depends on the direction and the wavelength.**




## 2.2 Stability of the AMUDIS

Prior to the outdoor measurements, the measuring device was calibrated according to Niedzwiedz et al. (2021) under temperature-controlled laboratory conditions.

During the measurements, the voltage of the three lamps inside of the used calibrating sphere (Ulbrichtsphere) (Niedzwiedz et al., 2021; Walker et al., 1987) fluctuated below +/- 0.001 A (Ampere) and +/- 0.016 A for each lamp. Consequently, the lamps were adequately stable for this investigation.


In total, 21 measurements were carried out for the calibration with timesteps of 1 minute. All measurements were corrected for dark current. Dark current refers to thermally generated electrons in a CCD sensor that can be detected in the measurement signal (Seckmeyer et al., 2010). To analyse the measurements, the ratios between each of the 21 measurements and the average value of all was determined.


The average value $M_{av}$ (f) of the 21 measurements $M_k$ (f) for each fibre (f) is defined by Eq. (1):

$$M_{av}(f) = \frac{1}{21}\sum_{k=1}^{21} M_k(f) \tag{1}$$

The ratio $R_{stability}$ (f) between each measurement and the mean value for each fibre (f) is given by Eq. (2):

$$R_{stability}(f) = \frac{M_k(f)}{M_{av}(f)}, k = 1, \dots, 21 \tag{2}$$

Figure 2 shows the calculated ratios for fibres 4, 28 and 90 in the wavelength range 470 - 600 nm. In these figures, orientation
lines (dashed) are marked at 0.97 and 1.03 (3 % deviation) and at 0.965 and 1.035 (3.5 % deviation).







**Figure 2: Calculated ratios between the 21 measurements and the mean value of the measurements for fibres 4 (a), 28 (b) and 90 (c)**
**based on the laboratory measurements. All ratios are in the range of 0.97-1.03 with outliers for fibre 28.**

Figure 2 illustrates that the fluctuations of the three selected fibres vary in the range between 0.97-1.03 with outliers for fibre
28.

These variations can be explained by the statistical noise of the instrument and the general stability of the device. Possible
influence of uncertainties inside the calibration sphere as described in Niedzwiedz et al. (2021) cannot be excluded.

## 2.3 Measurement settings for the different sky situations

All measurements for each situation were performed with the settings in table 2. The mean value and the standard deviation
of the temperature inside the air conditioning box for each analysed time-period are also listed in the table.




| Software setup | |
| --- | --- |
| Numbers of measurements for average | 10 |
| Detector temperature | -69°C |
| Exposure time | 1000 ms |
| Shutter | on |
| Camera | Pixis1024BRExcelon_12120513 |
| Temperature in air conditioning box | |
| 10.01. 11:52 – 12:01 UTC | 20.6°C (+/- 0.5°C) |
| 18.01. 11:52 – 12:01 UTC | 20.9°C (+/- 0.6°C) |
| 18.01. 13:17 – 13:26 UTC | 20.8°C (+/- 0.6°C) |

**Table 2: Overview of the settings and the temperature in the air conditioning box (mean value and standard deviation)**

**2.4 Parameters describing different sky situations**

For three different sky situations, the temporal change (R (t,λ,f)) was calculated for each of the three fibres (f), wavelengths (λ) and time (t), using Eq. (3):


$$R\ (t, \lambda, f) = \frac{M_x(t,\lambda,f)}{M_{cloudless}\ (\lambda,f)} \tag{3}$$

From $R$ (t,λ,f), the average value $R_{average}$ (λ,f) is computed using Eq. (4):

$R_{average}(\lambda, f) = \frac{1}{10}\sum_{k=1}^{10} R_k(t, \lambda, f)$ $\tag{4}$

The relative standard deviation (RSD) of the ratios is calculated according to (Niedzwiedz et al., 2022)
 with the standard deviation (σ(λ,f)) and the average value for each fibre, wavelength and situation:

$RSD(\lambda, f) = \frac{\sigma(\lambda,f)}{R_{average}(\lambda,f)}$ x100 % $\tag{5}$



The following table (Table 3) shows an overview of the acronyms and their meaning as well as the corresponding unit of the variables used.

| Acronym and units overview | | |
|---|---|---|
| Acronym | Name | Unit |
| R | Ratio | no unit |
| $R_{average}$ | Average of ratios | no unit |
| f | Fibre | no unit |
| $\lambda$ | Lambda | nm |
| t | Time | minute |
| RSD | Relative standard deviation | % |
| $M_x$ | Measurement at one time | counts |
| $M_{cloudless}$ | Measurement at cloudless conditions | counts |

**Table 3: Overview of the acronyms and their meaning which are used for the equations. The corresponding unit of the variable is**
**also listed.**

## 3. Results

### 3.1. Cloudless case (10 January 2024 11:52 – 12:01 UTC)

On 10 January 2024, there was a cloudless sky situation between 11:52 – 12:01 UTC. The following HSI image (figure 3)
shows an example of this sky condition at 11:52 UTC with the position and fields of view of the selected fibres for 480 nm
(chapter 2.1, Tobar Foster et al., 2021). Fibre 90 is therefore oriented to the north, fibre 28 to the northeast and fibre 4 to the
east. This alignment also applies to the other two wavelengths considered. Likewise, the position of the sun is marked with a
black cross.




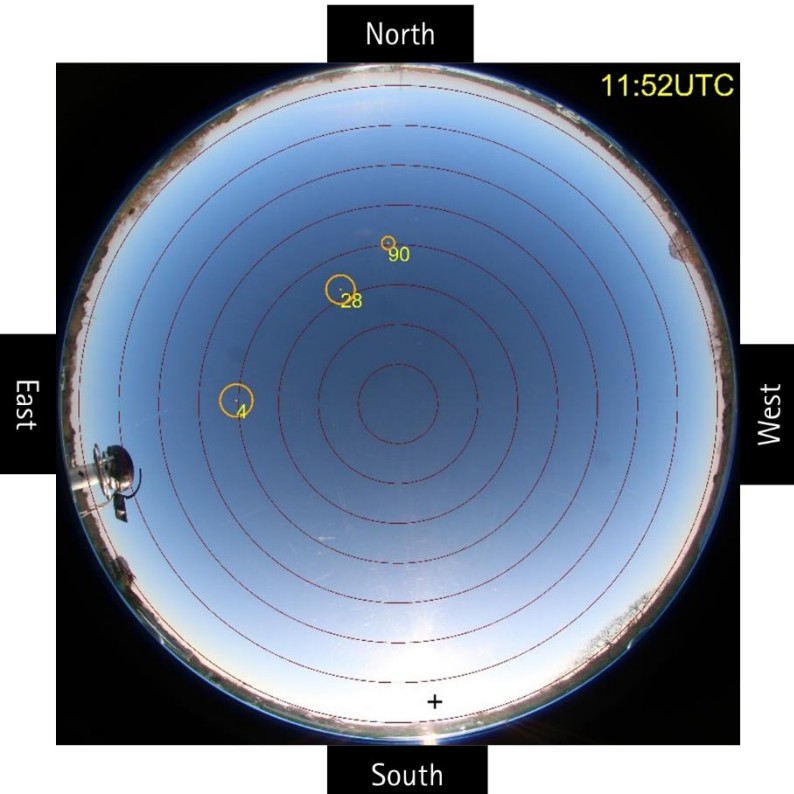

**Figure 3: Image of the HSI system from 10 January 2024 at 11:52 UTC. The positions of fibres 4, 28 and 90 and their FOV at 480 nm are included (based on Tobar Foster et al., 2021). The cross marks the position of the sun at this time. The dark red rings are used for orientation on the sky from the zenith to the horizon in 10.5° steps.**

The ratios were calculated with Eq. (3) based on the measurement at 11:52 UTC, which is defined in this case as the reference for the cloudless sky. Based on these ratios, the relative standard deviation (RSD) was calculated with Eq. (5).

The results of the three fibres and the respective wavelengths are illustrated in figure 4 and summarised in table 3.





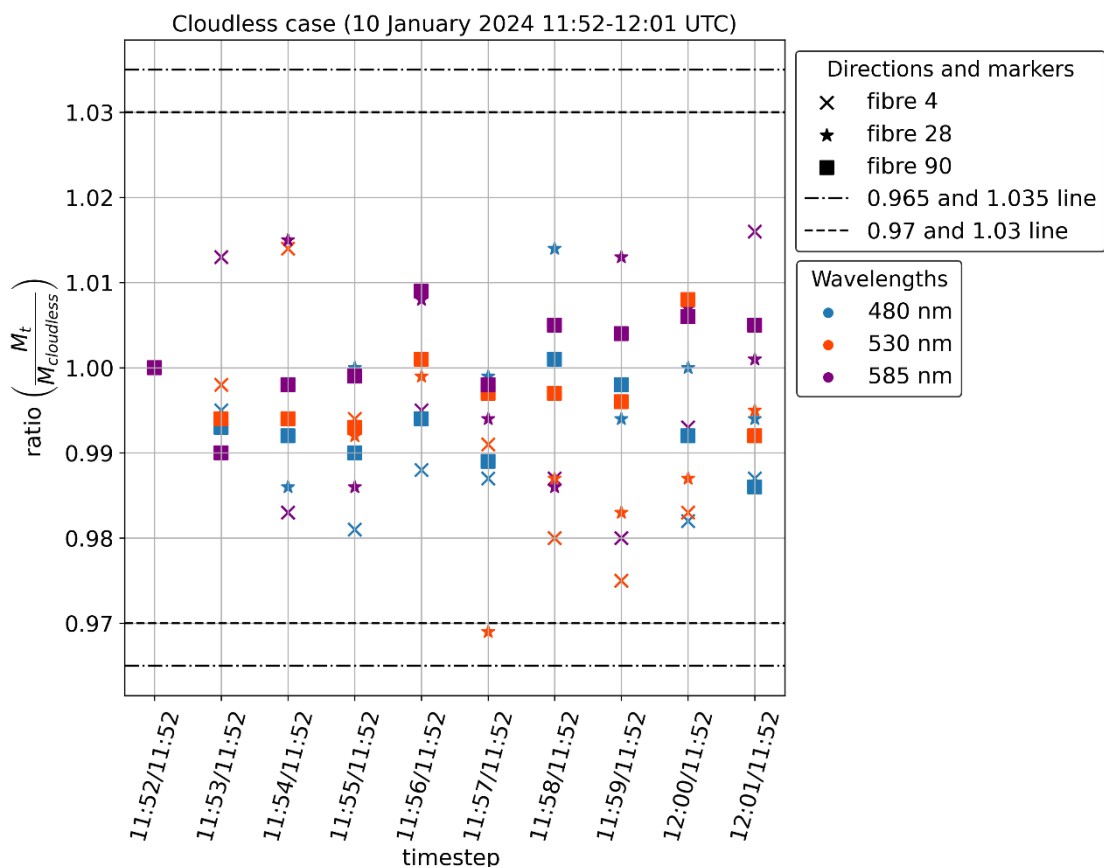

**Figure 4: Calculated ratios between the measurements and the cloudless situation defined for the cloudless case at 11:52 UTC (10 January 2024) for fibres 4, 28 ,90 and the three wavelengths 480 nm, 530 nm and 585 nm. The temporal changes are all in the range of 0.965 and 1.035.**

| Fibre 90 | 480 nm | 530 nm | 585 nm |
|---|---|---|---|
| min. Ratio | 0.986 | 0.992 | 0.99 |
| max. Ratio | 1.001 | 1.008 | 1.009 |
| average value | 0.99 | 1.0 | 1.0 |
| RSD [%] | 0.46 | 0.45 | 0.52 |
| Fibre 28 | 480 nm | 530 nm | 585 nm |
| min. Ratio | 0.986 | 0.969 | 0.986 |



| max. Ratio | 1.014 | 1.0 | 1.015 |
|---|---|---|---|
| average value | 1.0 | 0.99 | 1.0 |
| RSD [%] | 0.69 | 0.89 | 1.02 |
| Fibre 4 | 480 nm | 530 nm | 585 nm |
| min. Ratio | 0.975 | 0.975 | 0.98 |
| max. Ratio | 1.00 | 1.014 | 1.016 |
| average value | 0.986 | 0.99 | 1.0 |
| RSD [%] | 0.71 | 1.06 | 1.13 |

**Table 3: Overview of the minimum, maximum, average and RSD values between the measurements and the cloudless situation defined for the cloudless case at 11:52 UTC (10 January 2024) for fibres 4, 28 ,90 and the three wavelengths 480 nm, 530 nm and 585 nm.**


The results show that the ratios vary between 0.97-1.03 with outliers at 11:57 UTC (fibre 28, 530 nm). These changes may be explained by the device instability determined in chapter 2.2 (changes between 0.965 and 1.035). Nonetheless, a possible influence of aerosols or other atmospheric effects cannot be excluded.

The average value of the temporal changes for the cloudless sky is approximately 1 for all fibres and wavelengths. Consequently, there are no significant changes in the average during the measurement period.

The differences of Relative Standard Deviation (RSD) between the fibres are less than 1 %. Fibre 90 has the lowest RSD values (0.45 % - 0.5 %) compared to the other two fibres and therefore the lowest variation in the data. Fibre 4 in contrast, has the highest RSD values (0.82 % - 1.13 %). The reason could be the smaller FOV of fibre 90 (see chapter 2.1, Tobar Foster et

al., 2021), which means that this fibre detects a smaller area of the sky. In general, the differences of the RSD between these three fibres are less than 1% and therefore negligible. It should be noted that the RSD increases with wavelength for fibres 4 and 28. For fibre 90, as a contrast, these values fluctuate in a similar range over the three wavelengths. Compared to the laboratory measurements (see chapter 3.1), the variation of the data in relation to the mean value for fibres 28 and 90 decreases with increasing wavelength. This behaviour cannot be seen at fibre 4.

As previously mentioned, fluctuations within the range 0.965 – 1.035 are probably caused by device characteristics or the measurement setup. In this cloudless case, all ratios are within this range. It is assumed that all values lower than 0.965 or higher than 1.035 are attributable to clouds or aerosols (sections 3.2 and 3.3).




## 3.2 Broken cloud with low cloud cover case (18 January 2024 11:52–12:01 UTC)

On 18 January 2024 there was a cloudy sky situation between 11:52 – 12:01 UTC. Clouds moved from northwest to southeast. The following image (figure 5) shows the sky conditions for the whole timeframe with the same position and FOV of the
selected fibres for 480 nm (chapter 2.1, Tobar Foster et al., 2021) as in the cloudless case.

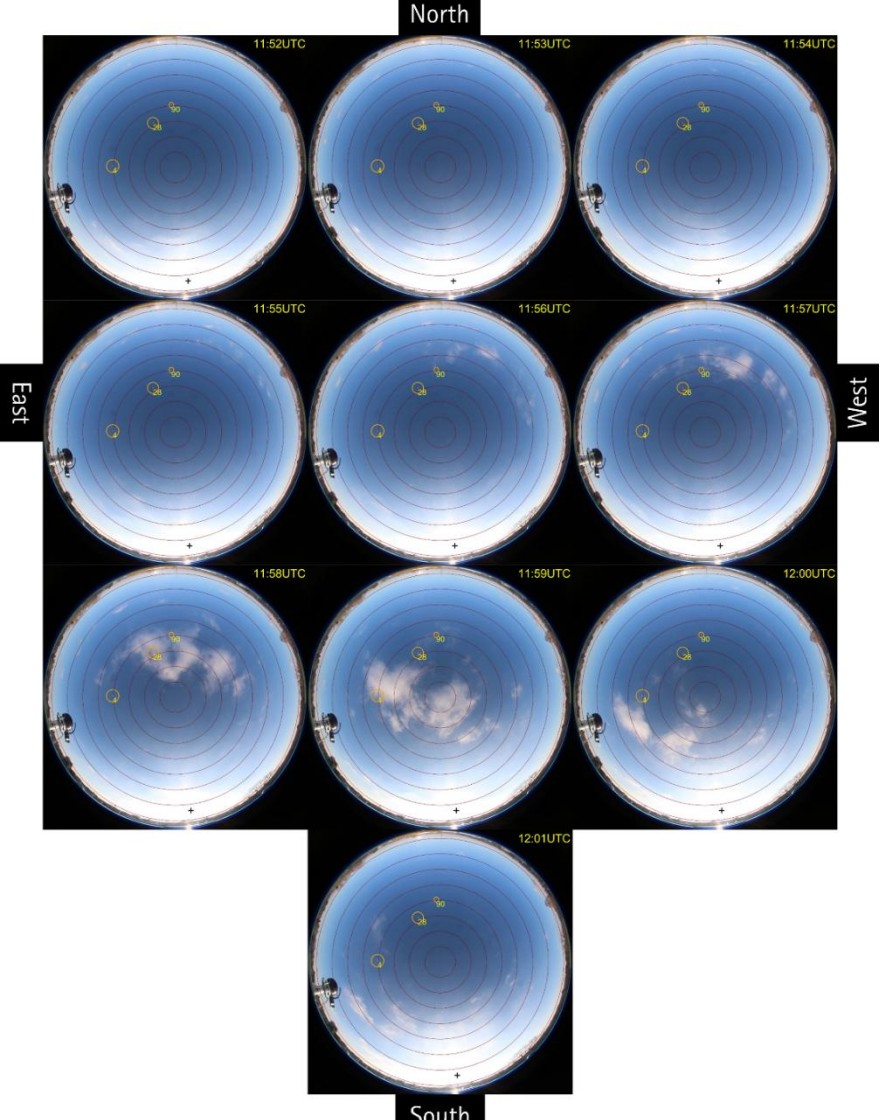

**Figure 5: Images of the HSI system from 18 January 2024 between 11:52 - 12:01 UTC. The positions of fibres 4, 28 and 90 and their respective FOV at 480 nm are included (based on Tobar Foster et al. (2021)). The cross marks the position of the sun at this time. The dark red rings are used for orientation dividing the sky from the zenith to the horizon in 10.5° steps. The clouds in the image**
**are visible from 11:55 UTC onwards.**




The temporal changes were calculated with Eq. (3) based on the measurement at 11:52 UTC, which was defined as the reference for the cloudless sky (section 3.2). Based on these ratios, the RSD was calculated by Eq. (5). The results of the three fibres and the respective wavelengths are illustrated in figure 6 and summarised in table 4.

The limit values determined as 0.965 and 1.035 (section 3.1) are used in the following graphs (dashed lines) to identify the influence of clouds and aerosols in the AMUDIS measurement. Values above the limit of 1.035 indicate an impact of clouds on the detected radiance. This higher threshold also excludes the possible influence of device instability shown in section 2.2.

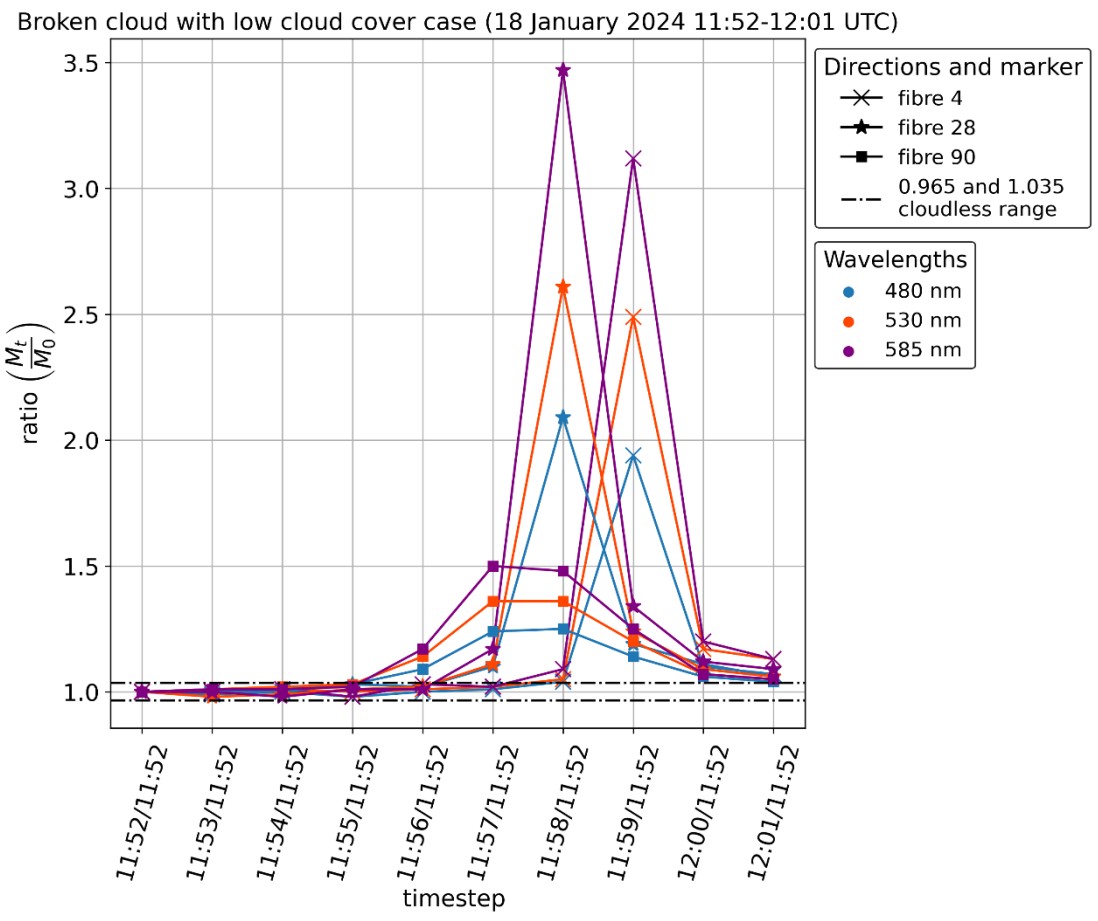

**Figure 6: Calculated ratios between the measurement in the period 11:52 - 12:01 UTC on 18 January 2024 and the cloudless situation defined for this period (11:52 UTC 18 January 2024) for the fibres 4, 28, 90 and the wavelengths 480, 530 and 585 nm. It can be noticed that from 11:56 UTC, the ratios increase above the cloudless limit (1.035) at different times (fibre 90 -> fibre 28 -> fibre 4) with maximum values of up to 3.47.**




| Fibre 90 | 480 nm | 530 nm | 585 nm |
|---|---|---|---|
| max. value (ratio) | 1.25 | 1.36 | 1.5 |
| average value (ratio) | 1.09 | 1.12 | 1.16 |
| RSD [%] | 8.15 | 11.73 | 15.87 |
| Timesteps (UTC) ratio > 1.035 | 11:56, 11:57, 11:58, 11:59, 12:00, 12:01 | | |
| Fibre 28 | 480 nm | 530 nm | 585 nm |
| max. value (ratio) | 2.09 | 2.61 | 3.47 |
| average value (ratio) | 1.16 | 1.21 | 1.32 |
| RSD [%] | 27.2 | 38.99 | 54.94 |
| Timesteps (UTC) ratio > 1.035 | 11:57, 11:58, 11:59, 12:00, 12:01 | | |
| Fibre 4 | 480 nm | 530 nm | 585 nm |
| max. value (ratio) | 1.94 | 2.49 | 3.12 |
| average value (ratio) | 1.11 | 1.19 | 1.26 |
| RSD [%] | 24.98 | 36.84 | 49.61 |
| Timesteps (UTC) ratio > 1.035 | 11:58, 11:59, 12:00, 12:01 | | |

**Table 4: Overview of the minimum, maximum, average and RSD values between the measurements and the cloudless situation**
**defined for the cloudless case at 11:52 UTC (18 January 2024) for fibres 4, 28 ,90 and the three wavelengths 480 nm, 530 nm and**
**585 nm.**

The calculated ratios are within the cloudless case limit values for all fibres between 11:52 and 11:55 UTC.

At 11:56 UTC, the deviations of fibre 90 rise above the value of 1.035 with a maximum at 11:57 UTC (1.36 (530 nm), 1.5

(585 nm)) and at 11:58 UTC (1.25 (480 nm), 1.36 (530 nm)). One minute later, the deviations of fibre 28 also rise above this

limit. The maximum values of this fibre are reached at 11:58 UTC (2.09 (480 nm), 2.61 (560 nm), 3.47 (585 nm)) for all three

wavelengths. Finally, the deviations of fibre 4 rise above the limit range at 11:58 UTC. The maximum values are reached at

11:59 UTC for all three wavelengths (1.94 (480 nm), 2.49 (530 nm), 3.12 (585 nm)). Except for fibre 90 (480 nm), all ratios

for all fibres are above the value of 1.035 until 12:01 UTC.

Considering the temporal progression and the exceeding of the cloudless limit value of 1.035, the movement of the clouds

from north to east can be identified in the ratios. Firstly, fibre 90 (11:56 UTC, northern orientation), following fibre 28 (11:57

UTC, northeastern orientation) and finally fibre 4 (11:58 UTC, eastern orientation) pass this limit.





Not only spatial-temporal deviations depending on the position of the sun and cloud in relation to the detecting fibre can be seen, but also the varying spectral influence of the cloud on the simultaneously measured radiances due to possible temporal

changes of the cloud position, properties (e.g. droplet size etc.) and their respective changes caused by reflecting and scattering processes. The possible influence of the different fields of view between the fibres also need to be considered [Tober Foster et al.2021.]

Similar to the maximum values of the ratios, fibre 90 also has the lowest mean values of the ratios (1.09 (480 nm), 1.12 (530

nm), 1.16 (585 nm)) and fibre 28 the highest (1.16 (480 nm), 1.21 (530 nm), 1.32 (585 nm)) of all three wavelengths. The mean values of fibre 4 differ in the ranges of the other two fibres (1.11 (480 nm), 1.19 (530 nm), 1.26 (585 nm)). Furthermore, the spatial, temporal and wavelength-dependent influence of clouds and aerosols on the radiance in relation to the fibre-position can be seen compared to the reference condition with cloudless sky (section 3.1).

Comparing the wavelength-dependent RSD of the three fibres, fibre 90 has the lowest values and therefore the lowest

deviations of the ratios, while fibre 28 reaches the highest values. Fibre 4 has similar values to fibre 28 and therefore a similar variation of the ratios due to different maximum values. In comparison to fibre 28 and 4, the wavelength-dependent RSD fluctuates less for fibre 90. In general, the RSD also shows a similar wavelength-dependence as the ratios and the mean values. According to section (2.1), this may be caused by a combination of the different wavelength-dependent field of views of the fibres [Tober Foster et al., 2021], the temporal and spatial sun – cloud radiation effects and the general time-varying cloud

properties in relation to the detecting fibre-position.

### 3.3. Broken cloud with higher cloud cover case (18 January 2024 13:17-13:26 UTC)

On 18 January 2024, the cloud cover was broken between 13:17 – 13:26 UTC. Clouds of different types moved from northwest

to southeast.

The following image (figure 7) shows the sky conditions for the whole timeframe with the same position and FOV of the selected fibres for 480 nm (chapter 2.1, Tobar Foster et al., 2021) as in the other two cases. The images show (see figure 7) that individual broken clouds are distributed all over the hemisphere.

In this case, two different measurements were defined as a reference for cloudless sky conditions. These measurements were used to calculate the spatial, temporal and spectral changes at different positions for simultaneous measurement-times. The measurement on 18 January 2024 at 13:17 UTC for fibre 4 and the measurement at 13:19 UTC for fibres 28 and 90 were used as the reference measurement. Equation (3) was used again to calculate the ratios and Eq. (5) was also used to calculate the RSD for the three fibres and the wavelengths. As in the other cloud case (chapter 3.2), the limit values (0.965 and 1.035) are



marked in the figures (dashed lines). Due to possible larger effects of sun-cloud and clouds-clouds interactions, scenarios below (< 0.965) as well as above (> 1.035) the limit values are analysed in the following.

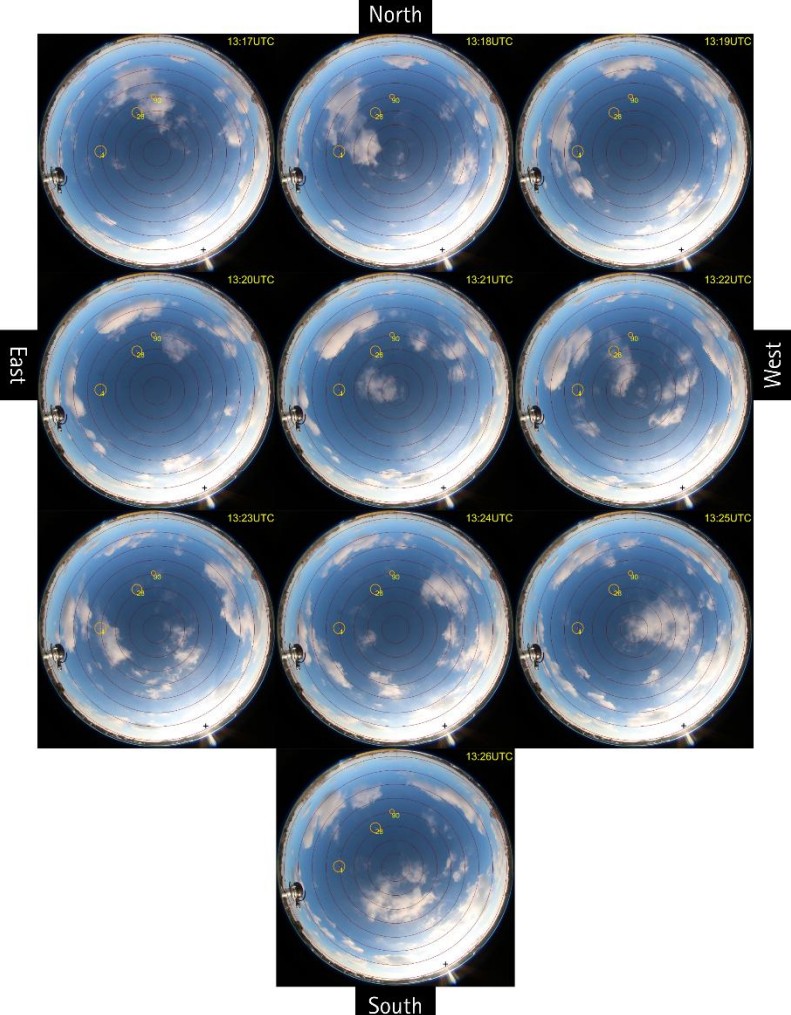

**Figure 7: Images of the HSI system from 18 January 2024 between 13:17 and 13:26 UTC. The positions of fibres 4, 28 and 90 and their FOV at 480 nm are included (based on Tobar Foster et al., 2021). The cross marks the position of the sun at this time. The dark red rings are used for orientation dividing the sky from the zenith to the horizon in 10.5° steps. Clouds are visible in different positions in all the images.**





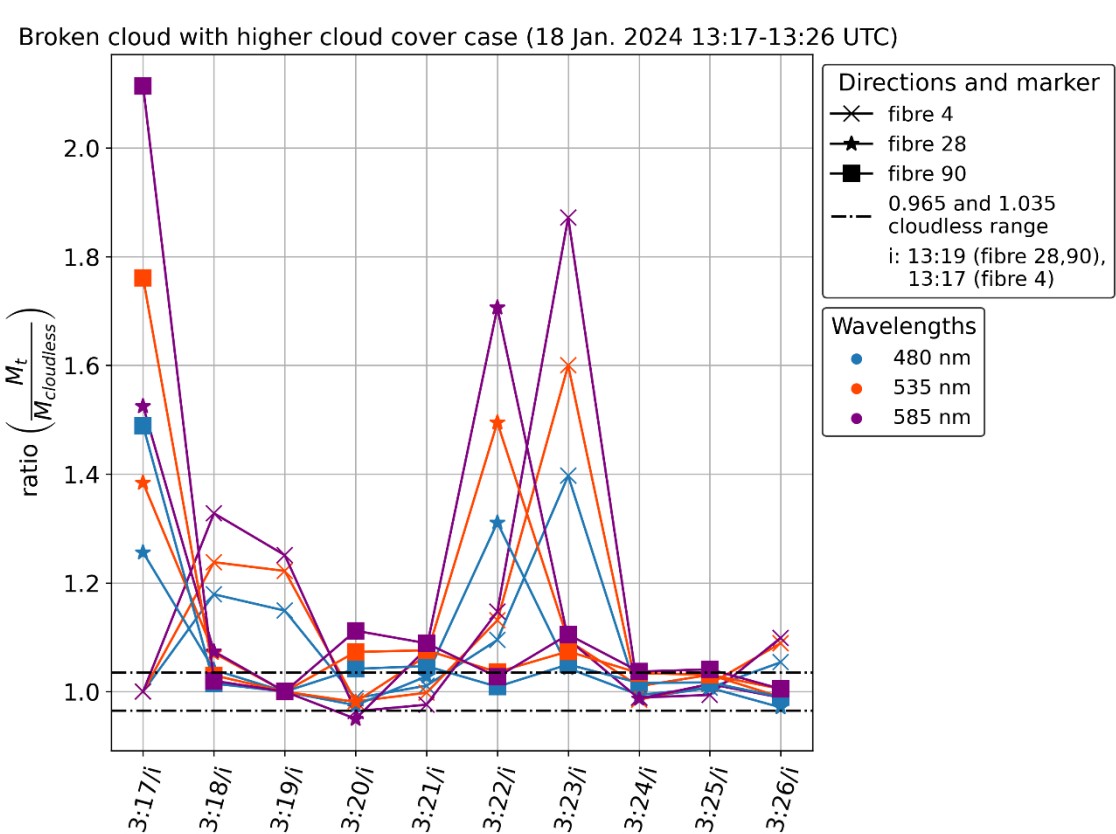

**Figure 8: Calculated ratios between the measurements and the cloudless situation defined for the cloudless case at 13:17 UTC (fibre 4) and 13:19 UTC (fibre 28, 90) (18 January 2024) for fibres 4, 28 ,90 and the three wavelengths 480 nm, 530 nm and 585 nm. It can be recognised that the ratios of the three fibres increase over the defined cloudless limit (1.035) at various times. The magnitude of these changes depends not only on the fibre and therefore on the direction, but also on the wavelength, with values of up to 2.11.**



| Fibre 90 | 480 nm | 530 nm | 585 nm |
|---|---|---|---|
| max. value (ratio) | 1.49 | 1.76 | 2.11 |
| average value (ratio) | 1.07 | 1.11 | 1.16 |
| RSD [%] | 13.25 | 19.55 | 27.77 |
| Timesteps (UTC) ratio > 1.035 | 13:17, 13:20, 13:21, 13:23 | 13:17, 13:20, 13:21, 13:22, 13:23 | 13:17, 13:20, 13:21, 13:22, 13:23, 13:24, 13:25 |
| **Fibre 28** | **480 nm** | **530 nm** | **585 nm** |
| max. value (ratio) | 1.31 | 1.5 | 1.71 |
| average value (ratio) | 1.06 | 1.11 | 1.14 |
| RSD [%] | 10.71 | 15.27 | 21.39 |
| Timesteps (UTC) ratio > 1.035 | 13:17, 13:18, 13:22, 13:23 | 13:17, 13:18, 13:21, 13:22, 13:23 | |
| Timesteps (UTC) ratio < 0.965 | – | – | 13:20 (0.95) |
| **Fibre 4** | **480 nm** | **530 nm** | **585 nm** |
| max. value (ratio) | 1.4 | 1.6 | 1.87 |
| average value (ratio) | 1.09 | 1.13 | 1.16 |
| RSD [%] | 11.28 | 16.2 | 22.78 |
| Timesteps (UTC) ratio > 1.035 | 13:18, 13:19, 13:22, 13:23, 13:26 | | |
| Timesteps (UTC) ratio < 0.965 | – | – | 13:20 (0.964) |

**Table 5: Overview of the minimum, maximum, average and RSD values between the measurements and the cloudless situation defined for the cloudless case at 13:17 UTC (18 January 2024) for fibre 4 and at 13:19 UTC for 28, 90 for the three wavelengths (480 nm, 530 nm, 585 nm).**

The calculated temporal changes for the three fibres are higher than the limit determined in chapter 3.1 (1.035) at different times. In addition, the temporal change for fibres 4 and 28 is lower than the cloudless limit (0.965) for one time step.




For example, the ratio of fibre 4 for all three wavelengths at 13:18 UTC, 13:19 UTC, 13:22, UTC 13:23 UTC and 13:26 UTC are above 1.035. In this analysed case, the ratios at 13:20 UTC for fibre 4 and fibre 28 at 585 nm are below 0.965. Different times result for fibre 28 and fibre 90 (see figure 8).

For the maximum values, it is shown that fibre 90 has the highest values in this case (13:17 UTC, 1.49 (480 nm), 1.76 (530 nm), 2.11 (585 nm)). Fibre 28 has the lowest values (13:22 UTC, 1.31 (480 nm), 1.5 (530 nm), 1.71 (585 nm)). Fibre 4 has slightly higher values than fibre 28 (13:23 UTC, 1.4 (480 nm), 1.6 (530 nm), 1.87 (585 nm). It should also be noted that for fibre 4 and fibre 28, the ratios are lower than the cloudless limit (0.965) at 13:20 UTC. In this case, the maximum values again occur in the same order as the movement direction of the clouds (fibre 90 -> fibre 28 -> fibre 4).

Not only spatial-temporal deviations depending on the position of the sun and cloud in relation to the detecting fibre can be seen, but also the varying spectral influence of the cloud on the simultaneously measured radiances due to possible temporal changes of the cloud position, properties (e.g. droplet size etc.) and their respective changes caused by reflecting and scattering processes. The possible influence of the different fields of views between the fibres also need to be considered [Tober Foster et al.2021].

Unlike the maximum values, the average values of the ratios for the three fibres are very close (fibre 90: 1.07 (480 nm), 1.11 (530 nm), 1.16 (585 nm); fibre 28: 1.06 (480 nm), 1.11 (530 nm), 1.14 (585 nm); fibre 4: 1.09 (480 nm), 1.13 (530 nm), 1.16 (585 nm)). Furthermore, the spatial, temporal and wavelength-dependent influence of clouds and aerosols on the radiance in relation to the fibre position can be seen compared to the reference condition with cloudless sky (section 3.1).

Comparing the wavelength-dependent RSD of the three fibres, fibre 90 has the highest values and therefore the highest deviations of the ratios. Fibres 4 and 28, however, have similar values with slight differences. In general, the RSD also shows a similar wavelength dependence as the ratios and the mean values. According to section (2.1), this may be caused by a combination of the different wavelength-dependent fields of view of the fibres (Tobar Foster et al., 2021), the temporal and spatial sun – cloud radiation effects and the general time-varying cloud properties in relation to the detecting fibre position.

### 4. Conclusion and Discussion Results


The results show that it is possible to use the AMUDIS to measure the influence of clouds on spectral radiance in different directions and at different wavelengths simultaneously. For this aim, three different situations for cloudless sky (10 January 2024) and two broken cloud with different cloud cover (both on 18 January 2024) were analysed. The initial analyses of these measurements suggests that the fluctuations in the ratios in the cloudless case between 0.969 and 1.016 (section 3.1) are currently within the device's uncertainties between 0.965 and 1.035 (section 2.2). Comparing the reference measurements performed for cloudless sky and the two broken cloud situations, it can be shown that the interactions between clouds lead to



significantly larger ratios than in the cloudless sky case and that the identified limit values are exceeded (maximum values of the ratios: broken cloud with lower cloud cover: 1.25 – 3.47, broken cloud with higher cloud cover: 1.31 - 2.11).

These higher ratios and the other calculated higher comparison parameters (average value of the ratios and their RSD) may be

attributable to interactions in the atmosphere between sun-clouds, clouds-clouds and aerosols effects, due to multiple reflections, scattering and absorption-processes at and inside of the clouds.

Furthermore, following conclusions can be derived for the two broken cloud cases. In the case where the cloud cover is lower, the variations between the three fibres are higher than in the case with higher cloud cover. A possible explanation is that the

higher number of clouds means that the solar radiation is scattered more often thus reducing the directional dependence slightly. It is also noticeable that although there is more fluctuation in the second broken cloud case, the RSD is not higher than in the first cloud case. In addition, in the second cloud case, the ratios of two fibres (4, 28) are below the cloudless limit at one time step (13:20 UTC). This could be due to the clouds passing directly in front of the sun at 13:19, 13:21 and from 13:24 UTC (visible on the HSI images), which may already have reduced the solar radiation previously.


Based on the analysed data and scenarios, it is shown how the spectral radiance can be investigated under the influence of clouds from different directions simultaneously using the AMUDIS. Even 3 different fibre orientations (2 of them close to each other from each other) show significant spectral and spatial variations of the radiance under the direct sun – cloud interaction effects.

In both analysed broken cloud cases, the values of the ratios indicate a spectral dependence (the greater the wavelength, the higher the ratio). This influence may also depend on the cloud type and the optical density of the clouds and is therefore not constant over time.

**Data and Code availability statement**

The data and the code that support the findings of this study are available upon request from the authors.

**Author contributions**

JeD and KW planned the measurement campaign jointly. The software required for the measuring device and for analysing the data was written by JeD and KW. JeD, KW, AN carried out the measurement campaign. Finally, JeD analysed the main data. The manuscript was written by AN and JD. All authors discussed the results and provided important input and corrections.



**Acknowledgements**

We thank Holger Schilke and Ullrich Meyer for their technical support. Further appreciation goes to Torben Meinert, Louisa Mundry, Stefanie Bloch, Arik Wilkens, Stefan Riechelmann, Michael Schrempf and Ralf Zuber for their support during the realization. We used DeepL software to assist us in the translation of the text from German into English.

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
