# Peer review of "Case study on the impact of moving broken clouds on the spectral radiance"

_EGUsphere, 2025_

## Author Comment (AC1)

Hannover, 19.08.2025

Dear Editorial Team,

Dear Reviewer,

Thank you very much for your efforts and for taking the time to read our paper "Case study on the impact of moving broken clouds on spectral radiance". Your comments and suggestions were helpful and will further improve the quality of the paper. We hope we were able to answer your comments satisfactorily. Below are our responses to your comments.

Kind regards

Jens Duffert (lead author)

**General comments:**

In my opinion, the development of the AMUDIS instrument would be of interest to the community and is worth publishing. However, I believe that the current manuscript needs substantial changes before it could be considered suitable for publication. The necessary changes include explaining the novelty and potential scientific benefits of the instrument and clearly explaining some important instrument characteristics and data interpretation issues. Please find my specific comments below.

**Major comments**

**1. Comment:**
The paper should clearly explain the novelty and benefits of the new instrument. The manuscript should discuss any scientific, engineering, practical, or financial benefits over existing instruments such as all-sky imagers and spectroradiometers. This includes discussing how exactly AMUDIS differs from MUDIS.

**Answer:**
We thank the reviewer for this comment. It enables us to sharpen our presentation of the novelty of our approach. The advantage of AMUDIS compared to other instruments is that it can perform simultaneous measurements from more than 140 different directions with high spectral resolution. This offers a significant advantage compared to scanning devices, which require several minutes for this type of measurement (for example Blumthaler M, Gröbner J, Huber M, Ambach W, 1996). During the measurement of the AMUDIS, the spectral radiance does nearly not change. This technique has been also realized with the prototype instrument MUDIS (Riechelmann et al, 2013). The major difference between AMUDIS and MUDIS is that the measurement range is larger (280–1700 nm (AMUDIS) compared to 250–600 nm (MUDIS)) and the number of directions is larger (140 direction with AMUDIS compared to 113 directions with MUDIS (Riechelmann et al., 2013)). Compared to MUDIS, which consists of a spectrometer and a camera, AMUDIS has three cameras and three spectrometers (one for each wavelength range).

Blumthaler M, Gröbner J, Huber M, Ambach W: Measuring spectral and spatial variations of UVA and UVB sky radiance. Geophys Res Lett 23(5):547–50, https://doi.org/10.1029/96GL00248, 1996.
Riechelmann S., Schrempf M., Seckmeyer G.: Simultaneous measurement of spectral sky radiance by a non-scanning multidirectional spectroradiometer (MUDIS), Measurement Science and Technology, 24(125501), 8, https://doi.org/10.1088/0957-0233/24/12/125501, 2013.

**2. Comment:**
The paper should discuss what specific cloud retrievals AMUDIS seeks to enable: optical thickness, particle size, or perhaps something else? If the targeted retrieval approach has been used in past studies, they should be referenced; otherwise, the concept of the envisioned retrieval approach should be outlined. The discussion should include explaining whether absolute calibration is needed for the envisioned retrievals, and if it is needed, the manuscript should outline plans or expected opportunities for absolute calibration.

**Answer:**
The AMUDIS measures spectral radiance and therefore does not directly measure cloud parameters such as optical thickness, particle sizes or similar. In future, it will probably be possible to retrieve various parameters from the measurements by inverse modelling, but this is not the focus of this study. The focus of this study is how the radiance changes relative to a starting point when one or more clouds move through. For this type of investigation, no absolute data are necessary, although the temporal variation caused by the measuring device and the setup must be known. To determine this, the calibration setup in the UK-100 has been published by Niedzwiedz et al., 2021. In future, absolutely calibrated measurement data will also be compared with other measuring instruments and radiative transfer simulations.

Niedzwiedz A., Duffert J., Tobar-Foster M., Quadflieg E., Seckmeyer G.: Laboratory calibration for multidirectional spectroradiometers, Measurement Science and Technology, https://doi.org/10.1088/1361-6501/abeb93, 2021.

**3. Comment:**
Since the last sentence of the abstract says "The AMUDIS offers another option for validating radiation transfer models", the paper should explain how one could validate radiative transfer models using AMUDIS data that, at least for the moment, lacks absolute calibration.

**Answer:**
Thank you for this comment, we changed the text to make clear that the validation of radiation transfer models is just a future option. The idea behind our statement has been that it is possible to do statistical comparisons of changes in radiance under cloudy conditions. In the future, we intend to use data with an absolute calibration.

**4. Comment:**
Section 2.1: Several important aspects of the AMUDIS instrument should be added to this section. Specifically:

**a)** What is the spectral resolution and the number of wavelengths within the three spectral ranges?

**Answer:**
The AMUDIS can measure between 280 and 1700 nm and is split into three ranges. These have been set by the manufacturer to:

UV: 280 – 390 nm with a resolution of 0.1 nm per pixel

VIS: 380 – 890 nm with a resolution of 0.5 nm per pixel

NIR: 880 – 1700 nm with a resolution of 1.3 nm per pixel

It is therefore theoretically possible to measure 1024 different wavelengths in the VIS range, as the camera chip has 1024 x 1024 pixels. In practice however, it is not possible to use all wavelengths due to various technical limitations for example the pixels at the chip edges show a higher noise level). The calibration is an ongoing issue of our investigations and we aim to improve it and deal actually with the limitations.

**b)** Why do the field-of-views have different sizes for different fibers (as shown in Fig. 1 and Table 1)?

**Answer:**

Up to now we have no satisfactory explanation for this feature, although we have asked several specialists for fibre optics. One hypothesis is that it may be caused by the manufacturing process of the input optics. This could possibly also be a fibre property, as these have a very small diameter of approx. 50 µm. A more comprehensive analysis has been presented by (Tobar Foster et al., 2021.

Tobar Foster, M., Weide, E.L., Niedzwiedz, A. Duffert, J., Seckmeyer, G.: Characterization of the angular response of a multi-directional spectroradiometer for measuring spectral radiance. EPJ Techn Instrum 8, 12 (2021). https://doi.org/10.1140/epjti/s40485-021-00069-4, 2021.

**c)** Why does it take a few seconds to take all the measurements, as mentioned in Line 70? Is there some sort of scanning or perhaps a rotating spectral filter involved?

**Answer:**

No, there the system does not employ any scanning or rotating filters. These measurements take a few seconds because we average 10 measurements, with an exposure time of 1000 ms for each measurement. The reason for averaging the measurements is to reduce noise. We have changed the manuscript accordingly.

**d)** What is the instrument structure like, and how does it enable a single camera to take measurements for 140 fibers? I recommend expanding the instrument description and adding a schematic drawing to show the key components of the system.

**Answer:**

The instrument layout is shown in the figure 1 (new graphic). The measuring instrument consists of an input optics with over 140 openings covered with glass plates, each containing three glass fibres (one for each wavelength range). These glass fibres are then bundled according to wavelength range and each bundle is connected to a spectrometer and camera specific to the wavelength range (the VIS range is shown as an example in this diagram). Each camera is connected to the computer and controlled by software developed at the institute.

[Figure]

Entrance optics of the AMUDIS with over 140 openings. Each opening contains three glass fibres, one for each wavelength range (UV, VIS, NIR).

Entranceoptic

VIS-Cam

Spectrometer

USB-Data cable

Spectrometer+ Camera

Glasfibres for :
UV: 280-390nm
VIS:380-890nm
NIR: 880-1700nm

PC

*Figure 1: Sketch of the AMUDIS Setup.*

The reason why a camera can perform simultaneous measurements from over 140 directions at the same time is that each individual direction represents a row of pixels. In other words, the information about the wavelengths is on the x-axis and the information about the directions is on the y-axis (figure 2).

[Figure]

*Figure 2: Example of a measurement from AMUDIS (direct data from the camera chip) on 10 January 2024 at 11:52:00 UTC. The xpixel axis contains the wavelength information and the pixel axis the direction information. Individual absorption lines are also visible.*

**5. Comment:**
Line 120: The words "was calibrated" should be replaced, as Section 2.1 only discusses characterizing the stability of the instrument, whereas calibration would mean that we determine what instrument output corresponds to what radiative quantity (such as radiance in units of W/m**2/sr/μm).

**Answer:**
Thank you for pointing this out. We will change the manuscript accordingly.

**6. Comment:**
Table 2: It should be clarified what it means that the shutter was "on".

**Answer:**
We will replace the term "on" with "active".

**7. Comment:**
Section 3.3 should discuss how clouds influencing nearby cloudless locations impact the combination/comparison of cloudy and cloudless data. This is important because cloudless areas are known to be affected by clouds through several mechanisms (ranging from increased light scattering by aerosols that swell in the extra-humid air near clouds to increased light

scattering in cloudless areas due to the extra illumination coming from nearby bright cloud sides)—whereas if the cloudless reference case is from a completely cloud free time period, this is not an issue (but then temporal variations in atmospheric properties can cause complications).

**Answer:**

Thank you for pointing this out. We are aware of the influence of the surrounding clouds. However, the focus of this paper is to show that it is possible to detect changes in the radiance caused by clouds with AMUDIS even in a broken cloud case. The ratios are probably in this case a mixture of the direct and indirect influence of clouds. However, it can be assumed that the influence is significantly greater when a cloud enters the FOV of the fibres directly than the indirect influence. The results in Figure 8 in the paper support this hypothesis. In the last few minutes, the ratios are within the range of the current device uncertainty, even though the clouds in the surrounding area are changing, and only when clouds are directly in the FOV the ratios are significantly higher. Another hypothesis could be that clouds directly in the sun's field of view have an influence. However, these hypotheses need to be investigated further.

**8. Comment:**

Line 101: In contrast to the text and Figure 3, Figure 1 shows Fiber 28 in the northwest segment (and not in the northeast segment); the discrepancy should be resolved. Also, the text should discuss somewhere why these three fibers were selected for analysis. It should also explain why the number of fibers is set to 140. Also, it should be explained what the overlapping circles represent near the bottom right and bottom left horizon lines in the right panel of Fig. 1, and why having some overlapping fibers is useful.

**Answer:**

Thank you for pointing this out. Unfortunately, an error has crept into Figure 1, so that fibre 28 is incorrectly shown in a north-westerly direction. These three fibres were selected to suit the two situations, as there were technical problems with other fibres. Additionally, more directions would have made the diagram less clear. We changed the text/figure to correct the error

**9. Comment:**

Section 3 should discuss the results not only in terms of fibers, but also in terms of relative azimuth angles and provide more thorough physical interpretation/explanations. Even basic intuitive explanations would help, for example pointing out that the high ratios occur because sunlit cloud sides are brighter than blue sky in between clouds. Following up on this, it would be helpful to also consider data from a fiber that views the shadowy (and not the sunlit) side of clouds: How do the observations behave for those fibers? The discussion of physics should also consider the physics behind the statement in Line 400: "the greater the wavelength, the higher the ratio" by pointing out that this is largely (or at least partly) due to the fact that scattering by air molecules and small aerosol particles is weaker at longer wavelengths. (This makes the sky appear blue and be darker at longer wavelengths.)

**Answer:**

This is a table with the zenith and azimuth angle of the three fibres. These data based on the data described generated by the measurements in Tobar Foster et al., 2021.

| | Fibre 4 | | Fibre 28 | | Fibre 90 | |
|---|---|---|---|---|---|---|
| | zenith angle | azimuth angle | zenith angle | azimuth angle | zenith angle | azimuth angle |
| 480 nm | 42.5 | 271.3 | 33.8 | 333.4 | 42.5 | 356.4 |
| 530 nm | 42.6 | 271.2 | 33.8 | 333.3 | 42.5 | 356.4 |
| 585 nm | 42.6 | 271.2 | 33.8 | 333.2 | 42.5 | 356.5 |

The data shows that fibre 4 and fibre 90 have a similar zenith angle but differ in their azimuth angle. Fibre 28 has a zenith angle that is approximately 10° smaller and lies between the two fibres in terms of azimuth angle (slightly closer to fibre 90). This means that this fibre 'looks' slightly more directly into the sky.

A possible explanation of these results is based on the following graphs. It should be noted that this paper only analyses a period of 10 minutes. Therefore, the following statements are initial hypotheses and must be taken with caution.

In the first graphic, this is for the "Broken cloud with low cloud cover case". This case is divided into 4 situations:

1. There is no cloud in the sky. The fluctuations in the measurements are caused by device properties and changes in the atmosphere (in this case, these are smaller than the fluctuations are probably caused by the device properties only.
2. A cloud enters the FOV of the two fibres. The situation is such that the constellation cloud – AMUDIS optics – sun is present (from left to right in the figure). Thus, the cloud does not directly obscure the sun. As can be seen in Figure 6 in the paper, the ratios become greater than 1 for all fibres and wavelengths. This could be the result of the cloud reflecting radiation to the input optics. The wavelength dependence of the results can be caused by the reflection of radiation at the cloud surface and their return path through the atmosphere (Rayleigh scattering). This has already been discussed in Kylling et al., 1997 for the UV range. This would therefore occur on the 'bright' side of the cloud.
3. The cloud is now directly above the optics and continues to reflect the radiation as in 2.
4. The cloud moves on and is now between the entrance optics and the direct sun beam. This means that they now shade the AMUDIS optics, causing the ratios to decrease again. Since the radiation is "modified" again by scattering on the cloud surface and thus travels a longer path through the atmosphere. This also changes the radiation because of Rayleigh scattering.

This reflection on the cloud surface can then of course be different on the "white" side or on the dark side of the cloud, as the albedo of the cloud is different.

[Figure]

*Figure 3: Sketch of the sun cloud interaction in the case with low cloud cover.*

This also happens in the second case presented, the 'Broken cloud with higher cloud cover case'. However, it must be noted that the radiation scattered by the individual clouds is further influenced or 'modified' by other clouds. This multiple scattering by different clouds also reduces the directional dependence (case 1 in the lower figure). Also interesting are clouds in front of the direct sun beam (case 2 in the figure below) and clouds in the FOV of the fibres. This is the case presented in the paper between 13:20 and 13:22 UTC.

[Figure]

*Figure 4: Sketch of the sun cloud interaction in the case with higher cloud cover.*

All of the hypotheses mentioned are likely to be influenced by the type of cloud (e.g. due to differences in optical thickness) and the position of the sun. A larger data set is probably required in order to provide more comprehensive interpretations. In some cases, the clouds in the examples are quite transparent, meaning that there is hardly any dark side. In general, it is considered that the light side reflects more and thus causes an increase in the measured radiation, while the dark side causes a decrease (depending on the relative position of the cloud to the measuring device/fibre).

Tobar Foster, M., Weide, E.L., Niedzwiedz, A., Duffert, J., Seckmeyer, G.: Characterization of the angular response of a multi-directional spectroradiometer for measuring spectral radiance. EPJ Techn Instrum 8, 12 (2021). https://doi.org/10.1140/epjti/s40485-021-00069-4, 2021.

Kylling, A., Albold, A., Seckmeyer G.:  Transmittance of a cloud is wavelength-dependent in the UV-range: Physical interpretation, Geophysical Research Letters Volume 24, Issue 4 pp. 397-400, https://doi.org/10.1029/97GL00111, 1997

**Minor comments**

**Comment:**
Lines 23-24: It is unclear what "source" refers to; this should be clarified, or the word should be deleted.

**Answer:**
Thank you for pointing this out. We will delete it.

**Comment:**
Line 24: I recommend adding "Arguably," between "influence" and "Spectral".

**Answer:**
Thank you for pointing that out. However, we would prefer to keep it as it is in the manuscript.

**Comment:**
Line 24: The abbreviation "cf." stands for the Latin word "confer", meaning "compare", which is not needed here as there is no comparison in the sentence.

**Answer:**
Thank you for pointing this out. We will delete it.

**Comment:**
Lines 24 and 435: This reference format is unfamiliar to me and should be clarified. Is "DIN" the last name of the author? If so, it should be changed to "Din", and the first name should also be provided.

**Answer:**
Thank you for pointing this out. DIN means "Deutsches Institut für Normung". We will adjust the citation in the manuscript.

**Comment:**
Line 27: I guess "radiation properties for" should be changed to "radiative properties when using", because the current wording is unclear.

**Answer:**
Thank you for pointing that out. We will change it.

**Comment:**
Line 34: The word "like" should be replaced by "such as".

**Answer:**
We will adapt in "such as".

**Comment:**
Line 38: Clouds are not parameters of the atmosphere.

**Answer:**

Thank you for pointing this out. We will remove clouds from this section.

**Comment:**

Line 56: I recommend adding "such as", "e.g.,", or "for example" right after the first opening parentheses.

**Answer:**

Thank you for pointing this out. We will add this.

**Comment:**

Line 60: As the meaning of "DFG approval: GZ: INST 187 / 555-1 FUGG" is unclear (this is not a reference format), this text should be deleted or clarified.

**Answer:**

Thank you for pointing this out. We will delete it.

**Comment:**

Line 60: It is unclear what is meant by "large-scale device"; this should be explained, or the wording should be changed.

**Answer:**

Thank you for pointing this out. We will change it to "with an increased number of directions and a greater spectral range".

**Comment:**

Lines 61-62: The words "the temporal uncertainties in measurements in different direction of the atmospheric variability" should be changed as this is confusing. For example, what is meant by "temporal uncertainties": The instrument does not keep track of when exactly it takes certain measurements? Also, what is meant by "different direction of the atmospheric variability": What is the direction of atmospheric variability?

**Answer:**

Thank you for pointing this out. We will change the sentence: "With the AMUDIS, it is possible to reduce the temporal uncertainties in measurements in different directions of atmospheric variability." to "With the AMUDIS, it is possible to reduce the time span between consecutive measurements in different directions of the variability of the radiance."

**Comment:**

Lines 106-110: The caption should have three parts: Top left, bottom left, and right. The current description of three images using four sections (Left, top, bottom, right) is confusing. Also, the acronym IMUK should be explained.

**Answer:**

Thank you for pointing that out. We will add it and write out the acronym IMUK (meaning "Institut für Meteorologie und Klimatologie") in full in the caption.

**Comment:**

Line 108: The word "objective" should be replaced as it has a different meaning than intended.

**Answer:**

Thank you for pointing that out. We will replace the word "objective" with "lens." This will change "fisheye objective" to "fisheye lens.

**Comment:**
Figures 1 and 3: I recommend flipping the sky images to put east to the right side and west to the left side, which is the usual orientation.

**Answer:**
Thank you for pointing that out, but we would prefer to keep it as it is, as in this case it is like looking at the sky at this point.

**Comment:**
Line 126: The text "In total, 21 measurements were carried out" should be modified to indicate that there were 21 measurements for each of the 140 views and each of the spectral ranges (if this is correct).

**Answer:**
Thank you for pointing that out. Yes, there are 21 measurements for the 140 different directions and spectral information. We will change the sentence to: "21 camera images were taken from which the directional and spectral information was retrieved."

**Comment:**
Equation (4): Over what time period were the 10 measurements spread out? (In other words, how far apart in time were the 10 measurements?) This becomes clear in later section, but as is, readers could get discouraged by not understanding this here. If preferred, even just a sentence saying that this will be discussed later could help.

**Answer:**
Thank you for pointing this out. We will add the information to the section.

**Comment:**
Line 398: The text in the parentheses "(2 of them close to each other from each other)" should be reworded.
**Answer:**
Thank you for pointing that out. We'll change this section to "2 of them with only 8.7 (zenith angle) and 23.3 (azimuth angle) degree difference."

---

## Author Comment (AC2)

Hannover, 19.08.2025

Dear Editorial Team,

Dear Reviewer,

Thank you very much for your efforts and for taking the time to read our paper "Case study on the impact of moving broken clouds on spectral radiance". Your comments and suggestions were helpful and will further improve the quality of the paper. We hope we were able to answer your comments satisfactorily. Below are our responses to your comments.

Kind regards

Jens Duffert (lead author)

**General comments:**

The manuscript presents a case study in which the Advanced MUltiDIrectional Spectralradiometer (AMUDIS) was used to measure spectral radiance in different directions under cloudy and cloud-free conditions on two different days. It was observed that the spectral radiance was by a factor up to 3 higher under cloudy conditions.

The availability of ground-based spectral radiance is – despite of their relevance for cloud-radiation interaction studies – very limited and thus this study is in my opinion highly important to the community and suited for publication in AMT.

The manuscript is in general quite well structured and - with some exceptions - clearly written. The literature has been carefully selected and cited. Graphics and tables are in general clear and the captions self-explanatory (but not always, see minor comments). The physical interpretations of the results could partly be improved. In addition, an outlook on further planned activities and capabilities of AMUDIS should be included in the conclusions. When the substance of the content is improved by giving more thorough physical explanations of the results, and the focus is sharpened towards a more original, better structured and formulated conclusion including an outlook, this study will be an interesting and valuable contribution to the atmospheric science community and is in my opinion suited for publication in AMT.

**Specific comments:**

**Section 2. Devices Materials and Methods.**

**Comment:**
- state the spectral resolution of AMUDIS (e.g. in line 70)

**Answer:**
AMUDIS can measure between 280 and 1700 nm and is split into three ranges set by the manufacturer. These are:

UV: 280 – 390 nm with a resolution of 0.1 nm per pixel
VIS: 380 – 890 nm with a resolution of 0.5 nm per pixel
NIR: 880 – 1700 nm with a resolution of 1.3 nm per pixel

**Comment:**
- what about stray light and potential correction? Can you comment on that?

**Answer:**
Currently, no stray light correction is used. However, for this paper only the visible range was used. In this range we did not observe any significant difference between the dark signal and those ranges with wavelengths where no signal is expected. Therefore we concluded there is no detectable stray light in the VIS range. We operate the measuring device in a light-tight climate box so that no external radiation can reach the spectrometer. We also shade the device from direct sunlight using a shadow band, which minimises overexposure in certain directions.

**Comment:**
- Why is the FOV different for different fibers? Production and alignment of the fibers?

**Answer:**
Up to now we have no satisfactory explanation for this feature, although we have asked several specialists for fibre optics. One hypothesis is that it may be caused by the manufacturing process of the input optics. This could possibly also be a fibre property, as these have a very small diameter of approx. 50 μm. A more comprehensive analysis has been presented by (Tobar Foster et al., 2021.

Tobar Foster, M., Weide, E.L., Niedzwiedz, A. Duffert, J., Seckmeyer, G.: Characterization of the angular response of a multi-directional spectroradiometer for measuring spectral radiance. EPJ Techn Instrum 8, 12 (2021). https://doi.org/10.1140/epjti/s40485-021-00069-4, 2021.

**Comment:**
- calibration: I am confused: the method of Niedzwiedz et al. (2021) was used to calibrate AMUDIS (line 120). This is an absolute calibration method. However, you did not point out the absolute nature of the calibration but highlighted its stability only. Later you used counts for the individual measurements instead of $W/m^2/sr/\mu m$ what is in principle appropriate because you calculated ratios. Nevertheless, you should clearly state if you have calibrated AMUDIS absolutely, especially since you highlighted in the abstract that AMUDIS can be used for the validation of RTM, which requires an absolute calibration.

**Answer:**
Sorry for the confusion and thanks for pointing that out. We checked how stable the measuring device is by using the UK-100 mentioned in Niedzwiedz et al. 2021 and limited the noise to a variation of +/- 3.5%. The cited paper is intended to show that the UK-100 itself is stable and it can be used for this investigation and also later for absolut calibration. The count values are then used to calculate the relative changes. The note about the RTM was intended to provide some perspective. We changed the text accordingly to avoid the confusion.

Niedzwiedz A., Duffert J., Tobar-Foster M., Quadflieg E., Seckmeyer G.: Laboratory calibration for multidirectional spectroradiometers, Measurement Science and Technology, https://doi.org/10.1088/1361-6501/abeb93, 2021.

**Section 3: Results**

**Comment:**
- why have you chosen the three fibers 4, 28 and 90 for analysis? Is it for these particular cloud scenes? Would different directions may be more suitable for these specific cloudy scenes?

**Answer:**
We selected these three fibres because they correspond well to the trajectory of the cloud and think this selection is sufficient for a first case study.

**Comment:**
- you pointed out a potential impact of aerosols in the cloud-free scene (line 224). Is there any aerosol information available (AOD, single scattering albedo...), e.g. from a nearby AERONET site?

**Answer:**
Unfortunately, no parallel aerosol measurements were taken, so we do not have this data available. However, measurements using various measuring instruments are planned as part of the C3SAR project (Cloud 3D Structure And Radiation, https://c3sar.de).

**Comment:**
- the physical interpretations of the observed features are sometimes limited, e.g. what is the physical cause of individual readings to be lower than the uncertainty limit of the cloud-free reference (line 350)? Or what is the physical explanation for higher ratios for longer wavelengths (line 400)? Try to give a more concise and physically-based interpretation for such cases

**Answer:**
A first attempt to explain these results can be seen in the following graphs. It should be noted that this paper only analyses a period of 10 minutes. Therefore, the following statements are initial hypotheses and must be taken with caution.

In the first graphic, this is for the "Broken cloud with low cloud cover case". This case is divided into 4 situations:

1. There is no cloud in the sky. The fluctuations in the measurements are caused by device properties and changes in the atmosphere (in this case, these are smaller than the fluctuations are probably caused by the device properties only.
2. A cloud enters the FOV of the two fibres. The situation is such that the constellation cloud – AMUDIS optics – sun is present (from left to right in the figure). Thus, the cloud does not directly obscure the sun. As can be seen in Figure 6 in the paper, the ratios become greater than 1 for all fibres and wavelengths. This could be the result of the cloud reflecting radiation to the input optics. The wavelength dependence of the results can be caused by the reflection of radiation at the cloud surface and their return path through the atmosphere (Rayleigh scattering). This has already been discussed in Kylling et al., 1997 for the UV range. This would therefore occur on the 'bright' side of the cloud.
3. The cloud is now directly above the optics and continues to reflect the radiation as in 2.
4. The cloud moves on and is now between the entrance optics and the direct sun beam. This means that they now shade the AMUDIS optics, causing the ratios to decrease again. Since the radiation is "modified" again by scattering on the cloud surface and thus

travels a longer path through the atmosphere. This also changes the radiation because of Rayleigh scattering.

This reflection on the cloud surface can then of course be different on the "white" side or on the dark side of the cloud, as the albedo of the cloud is different.

[Figure]

*Figure 1: Sketch of the sun cloud interaction in the case with low cloud cover.*

This also happens in the second case presented, the 'Broken cloud with higher cloud cover case'. However, it must be noted that the radiation scattered by the individual clouds is further influenced or 'modified' by other clouds. This multiple scattering by different clouds also reduces the directional dependence (case 1 in the lower figure). Also interesting are clouds in front of the direct sun beam (case 2 in the figure below) and clouds in the FOV of the fibres. This is the case presented in the paper between 13:20 and 13:22 UTC.

[Figure]

*Figure 2: Sketch of the sun cloud interaction in the case with higher cloud cover.*

All of the hypotheses mentioned are likely to be influenced by the type of cloud (e.g. due to differences in optical thickness) and the position of the sun. A larger data set is probably required in order to provide more comprehensive interpretations. In some cases, the clouds in the examples are quite transparent, meaning that there is hardly any dark side. In general, it is

considered that the light side reflects more and thus causes an increase in the measured radiation, while the dark side causes a decrease (depending on the relative position of the cloud to the measuring device/fibre).

Kylling, A., Albold, A., Seckmeyer G.: Transmittance of a cloud is wavelength-dependent in the UV-range: Physical interpretation, Geophysical Research Letters Volume 24, Issue 4 pp. 397-400, https://doi.org/10.1029/97GL00111, 1997

**Section 4: Conclusion and Discussion results**

**Comment:**
I propose to include an outlook: what are your plans for AMUDIS in terms of technical activities and analysis, for instance:

- implementation of a stray-light correction (if necessary)?

- absolute calibration (if not yet conducted, see my comment above)

- are there any plans/possibilities to retrieve some macro-/microphysical aerosol and/or cloud properties from spectral radiance observations of AMUDIS? If yes, which ones?

- quantification of 3D radiative effects?

- validation of RTM

**Answer:**
In future, it is possible to calculate not only relative changes but also absolute changes using the absolute calibration described in the paper of Niedzwiedz et.al. In addition we intend to compare AMUDIS with other measuring devices. Further radiation effects are to be investigated as part of the C3SAR project (Cloud 3D Structure And Radiation, https://c3sar.de/). AMUDIS will there also be part of the large campaign planned for summer 2026. The calculated absolute radiance will be compare to the results of RTMs, at least in a statistical sense.

**Minor comments**

**Comment:**
Line 15: may insert "...the temporal variations of the spectral radiance was calculated..."
**Answer:**
Thank you for pointing this out. We will change it.

**Comment:**
Line 19: Validation of RTM: I would rather put this in the outlook. In addition, an absolute calibration of AMUDIS is required (see my comments above)
**Answer:**
Thank you for pointing this out. We will delete it here and add it to the outlook as described above.

**Comment:**
Line 23: delete "source"
**Answer:**
Thank you for pointing this out. We will delete it.

**Comment:**
Line 27: cloud and radiative properties
**Answer:**
Thank you for pointing this out. We will change it.

**Comment:**
Lines 33/34: May rephrase this sentence, e.g.: "Solar radiance at the surface can be observed using various measurement systems, such as:"
**Answer:**
Thank you for pointing this out. We will change it.

**Comment:**
Line 38: May rather use "constituents" than "parameters"
**Answer:**
Thank you for pointing that out. We will edit the sentence in lines 38 and 39

**Comment:**
Line 39: "...such as clouds, aerosols may not be detected..."
**Answer:**
Thank you for pointing this out. We will change the sentence with the clouds in and remove the clouds.

**Comment:**
Lines 46/47: may rephrase this sentence, e.g.: "...(HIS) systems, although they are mainly used to detect clouds, calculate the total cloud cover, classify clouds and study their radiative effects"

**Answer:**

Thank you for pointing this out. However, we would leave this sentence as it is.

**Comment:**

Line 51: Due to their automated…

**Answer:**

Thank you for pointing this out. We will change it.

**Comment:**

Line 56: May rephrase, e.g.: "Non-scanning multidirectional spectroradiometers such as the multidirectional spectral radiometer (MUDIS) and the advanced multidirectional spectral radiometer (AMUDIS)…"

**Answer:**

Thank you for the hint, we will change it so that we write in example after the first bracket.

**Comment:**

Line 60: What does "large-scale device" mean?

**Answer:**

Thank you for pointing this out. What we mean is that AMUDIS has a larger wavelength measurement range of 280–1700 nm than MUDIS, for example. We will adjust the wording accordingly.

**Comment:**

Line 60: I would delete the DFG approval

**Answer:**

Thank you for the hint. We will remove the DFG approval.

**Comment:**

Line 62: "temporal uncertainties in measurements in different direction of the atmospheric variability" sounds strange to me. You may mean that with AMUDIS you can observe the spatial and temporal variability of the atmospheric constituents and properties or similar, I guess.

**Answer:**

Thank you for pointing this out. We will change the sentence: "With the AMUDIS, it is possible to reduce the temporal uncertainties in measurements in different directions of atmospheric variability." to "With the AMUDIS, it is possible to reduce the time span between consecutive measurements in different directions of the variability of the radiance."

**Comment:**

Line 71: "…as described in Seckmeyer et al. (2018 and Tobar Foster et al. (2021), it is based on…"

**Answer:**

Thank you for pointing this out. We will change it.

**Comment:**

Line 87: may use "…the light is detected by three CCD image sensors"

**Answer:**

Thank you for pointing this out. We will change it.

**Comment:**

Lines 106-110: Caption of Figure 1: Use "upper left, bottom left and right" for the description of the three images. Define IMUK

**Answer:**

Thank you for pointing that out. We will add it and write out the acronym IMUK (meaning "Institut für Meteorologie und Klimatologie") in full in the caption.

**Comment:**
Line 150: May add "Thus, an instrumental uncertainty of 3.5 % is assumed " or similar
**Answer:**
Thank you for pointing this out. We will change it.

**Comment:**
Table 2: Shutter on: Replace "on" by "open"
**Answer:**
Thank you for pointing this out. We will change it.

**Comment:**
Line 234: May replace "at" by "for"
**Answer:**
Thank you for pointing this out. We will change it.

**Comment:**
Line 253: Isn't it section 3.1 instead of section 3.2?
**Answer:**
Thank you for pointing this out. We will take it out.

**Comment:**
Line 269 and Line 345: Table 4 and Table 5: The minimum of the ratio is not indicated.
**Answer:**
Thank you for pointing this out. We did not list the minimum ratios in the tables because they are all within the +/- 3.5% limit (except for one point in time in Section 3.3) and therefore do not offer any direct added value.